# Analysis of the Scale Effect and Temporal Stability of Groundwater in a Large Irrigation District in Northwest China

Ziyi Zan [1], Weifeng Yue [1,*], Hangzheng Zhao [1], Changming Cao [1], Fengyan Wu [2], Peirong Lin [2] and Jin Wu [3]

1 College of Water Sciences, Beijing Normal University, Beijing 100875, China
2 Hubei Water Resources and Hydropower Science and Technology Promotion Center, Hubei Water Resources Research Institute, Wuhan 430070, China
3 Faculty of Architecture, Civil and Transportation Engineering, Beijing University of Technology, Beijing 100124, China
* Correspondence: yuewf@bnu.edu.cn

**Abstract:** The depth to groundwater table (DGT) and the stability sites of groundwater were closely related parameters in groundwater research. Controlling the DGT and identifying stability sites of DGT were of great significance to prevent soil salinization and improve groundwater monitoring. In this study, using DGT data from the Hetao Irrigation District (HID) from 1991 to 2015, combined with spatial interpolation and coefficient-of-variation methods, this study explored the spatiotemporal variation characteristics and scale-effect problems of DGT from four hierarchical scales: the irrigation district, irrigation subdistrict, main canal, and branch canal. The Spearman correlation coefficient, average relative difference, and standard deviation were also used to further clarify the characteristics of groundwater time stability and its periodic variation rule. The results indicated that the spatiotemporal variation in DGT in the HID, and showed moderate variation characteristics, consistent with scale-effect features, which was deeply influenced by the regional climate and human activities. The DGT in the HID showed different temporal stabilities before and after 2000 caused by the application of Water-saving practices (WSPs). The stability sites were not entirely the same in different years or time periods, but they were all at the moderate DGT level in the HID. The results of this study can provide more insights for improving soil salinization and groundwater monitoring and provide more information for agricultural water-use efficiency and management.

**Keywords:** depth to groundwater table; spatiotemporal variability; scale effect; time stability; Hetao irrigation district





## 1. Introduction

The distribution of groundwater exploitation in China is extremely uneven, mainly concentrated in the northern region, especially in arid and semiarid areas, where it serves as the most important irrigation water source [1], accounting for approximately 70% of agricultural water use [2]. However, caused by strong evaporation and little rainfall in the Hetao Irrigation District (HID), a large irrigation district in Northwest China, as well as shallow depth to groundwater table (DGT), the problem of soil salinization is particularly serious in this area [3]. It is very important to protect the soil from the impact of salinization to maintain ecological balance, guarantee agricultural production, and improve land-use efficiency. Moreover, soil salinization can also have negative impacts on the stability and diversity of ecosystems [4]. Therefore, in irrigated agricultural ecosystems, it is of great significance to reduce the impact of soil salinization. However, DGT is one of the main indicators used to determine whether irrigation in a certain area will cause soil salinization [5]. Suitable DGT can maintain the ecological functions of the soil—plant—atmosphere continuum during the dry season [6]. Therefore, by understanding the spatial and temporal distribution characteristics of DGT, a scientific basis can be provided for the improvement of the HID and the sustainable development of agricultural production in

the Yellow River Basin [7]. This can also help to improve agricultural productivity and the quality of the water–soil ecological environment.

Scale is a specialized description of time and space that has different meanings in different fields [8]. In the early stage, there were many studies on the spatial variability of soil, which showed that the greater the scale was, the stronger the variability revealed, and there was a clear positive correlation between the two indicators [9]. Subsequently, some researchers proposed that the change in variability with the change in scale was caused by the scale effect and defined this property as scale dependence [10]. It can help us understand how certain features of the object being studied change with variations in the measurement scale or the area of the region or how they are affected by resolution [11]. By studying these dependencies, we can better understand the fundamental characteristics and patterns of the object being investigated. To date, many scholars have studied soil properties [12,13], the spatiotemporal evolutionary characteristics of soil water [14], and their scale effects. Researchers have also explored the spatiotemporal variability of soil salinity at different scales. For example, Jiang [15] investigated the spatial variability of soil salinity in the Yanqi Basin of Xinjiang at the profile scale, field scale, and irrigation district scale and discussed the optimal sampling scheme for the Yanqi Basin. Ren et al. [16] studied the changes in soil salinity in the HID at the field scale, canal scale, and regional scale. Given the insufficiency of previous research, it is necessary to analyze and discuss the scale effect of the spatiotemporal variability of DGT to further explore the regular characteristics of DGT. It is also vital to investigate the effects of different spatiotemporal monitoring scales on the variability of groundwater.

There have been many achievements in spatiotemporal variability, mainly through geostatistics and model simulation. In recent years, due to the powerful data analysis functions of Geographical Information System (GIS) and the accurate interpolation of geostatistics, the combination of geostatistics and GIS technology has provided a better exploration of spatiotemporal variability. Xiao et al. [17] used GIS software to analyze the spatiotemporal variability of groundwater level from 2001 to 2013 and demonstrated that the Kriging method can better fit the spatiotemporal variability characteristics of groundwater. Lu et al. [18] studied the spatial distribution characteristics and dynamic patterns of DGT on the Northeast China Plain and the North China Plain and noted that DGT was influenced by climate, topography, geology, and human activities. Yue et al. [19] combined the empirical orthogonal function (EOF) method and grey correlation analysis to quantitatively showed that meteorological factor was the main reason for spatial variability in the Yichang irrigation subdistrict of the HID. Dash et al. [20] indicated that terrain and soil texture were the main factors driving spatial variability of groundwater. However, recent studies have found that these driving factors exhibited different behaviors under the influence of anthropogenic activities [21]. Singh et al. [22] conducted a spatiotemporal survey of groundwater level fluctuations in India and proposed the joint utilization of ground and surface water, as well as the adoption of artificial recharge projects, to alleviate the sharp decline in groundwater levels. Varouchakis et al. [23,24] compared with traditional geostatistical methods, the use of Gaussian process method and spatiotemporal regression Kriging method for spatiotemporal modeling of groundwater level can better evaluate the spatial distribution of groundwater level in the Mediterranean basin.

The spatiotemporal variability of soil is characterized by temporal stability. This concept was originally defined by Vachaud as a type of time-invariant correlation between spatial location and classical statistical parameter values [25]. Zhao et al. [26] used a grey time series combination model to explore the fluctuation period of the groundwater level. Ran et al. [27] found that there was strong temporal stability in the spatial pattern of groundwater. He et al. [28] showed that the spatial pattern of water quality can be maintained in the short term but not in the long term. Xu et al. [29] studied the conductivity of groundwater in the Luohui Canal irrigation area and found that it exhibited temporal stability and periodic regularity. By using the monitoring data from representative wells, the irrigation time can be accurately determined. Wang et al. [30] compared the groundwater

data from the representative sites with the remote sensing data in the contemporary Yellow River Delta, and the results showed that representative sites can be used to verify DGT based on satellite data instead of the regional average value. Therefore, evaluating the temporal stability of DGT is helpful to optimize the groundwater monitoring network.

As mentioned earlier, reasonable control of DGT is of great significance for improving and preventing saline–alkali land. Currently, there have been many studies on irrigation water use in the HID. The main focus has been on the overall study of preventing and controlling soil salinization and the spatiotemporal variation characteristics of DGT. The regional scale effect of DGT variation characteristics has not been well explored, especially in multi-scale studies from irrigation districts, subdistricts, and main canals to branch canals, which can better explore the spatial structural change characteristics. At present, there is also a lack of research on the temporal stability and periodic changes in DGT. The exploration of temporal stability can not only improve the effective utilization of resources but can also provide a strong theoretical and practical basis for better agricultural production.

The main objectives of this study are as follows: (1) to analyse and explore the spatiotemporal patterns and changes in DGT from different regional scales; (2) to elucidate the temporal stability characteristics of DGT in the HID as well as the impact of water-saving practices (WSPs) on it; and (3) to determine the time-stable monitoring sites at different regional scales. The results of this study can provide more insights for controlling soil salinization, support the optimization of groundwater monitoring networks, and provide more information for the management of groundwater resources in China in the future.

## 2. Materials and Methods

### 2.1. Study Area

The HID ($40°20'\sim41°18'$ N, $106°20'\sim109°20'$ E) is located in the upstream of the Yellow River Basin which is primarily focused on agriculture (Figure 1). The region has a typical arid and semiarid continental climate, with an average annual maximum temperature of 27.19 °C and a minimum temperature of −14.40 °C, occurring in July and January, respectively. The annual average precipitation is approximately 146.65 mm, with more than 70% occurring from July to September, and the pan evaporation is 2280 mm. The HID mainly depends on the water diverted from the Yellow River to irrigate crops, and the average annual water diversion is approximately 3.743 to 5.203 billion m$^3$ per year. The area of the HID is about 11,073 km$^2$, with an irrigated area of 5740 km$^2$ that diverting water to irrigate from the Yellow River. This study divided the HID into four scales; except for the HID, the other three scales were the Yichang Irrigation Subdistrict (YCIS), the Yihe Main Canal Area (YHMC), and the Branch Canal Area (BC). The respective areas are 3321.05 km$^2$, 667.3 km$^2$, and 121 km$^2$.

### 2.2. Data

The data mainly include monthly groundwater monitoring data from 1991 to 2015 in 206 observation wells in the HID. The DGT and water intake data were provided by the Hetao Experimental Station in Inner Mongolia, while meteorological data such as precipitation and evaporation were provided by the China Meteorological Network (http://data.cma.cn, accessed on 2 March 2022). Based on the observation data of meteorological stations from 1991 to 2015, the potential evapotranspiration of the HID was calculated using the Penman–Monteith formula.

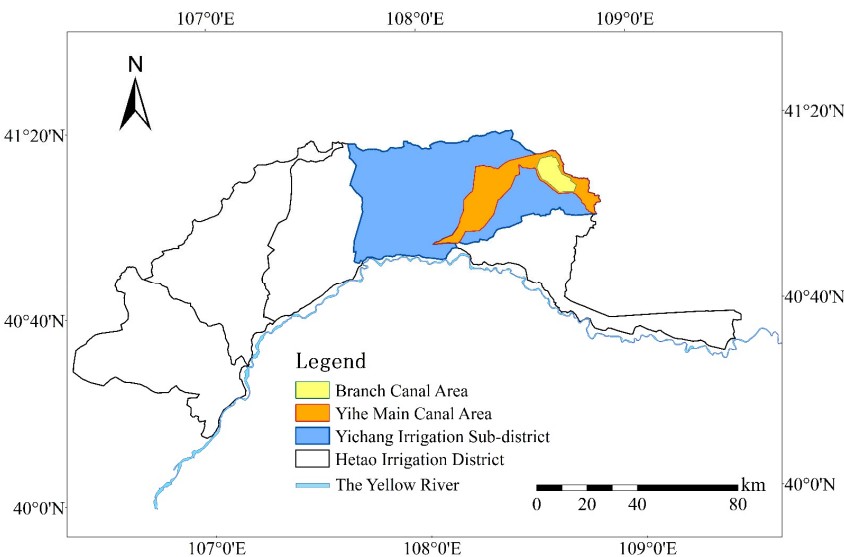

**Figure 1.** Overview map of the study area.

*2.3. Methods*

The relative difference method was used to measure the deviation degree of the object from the average value. The project used the average relative difference and its standard deviation to reflect the temporal stability of the DGT. Therefore, the principle of "when the average relative deviation is close to zero, the standard deviation is small" was adopted to select stable well sites on behalf of the average working condition. The relative difference is defined as [30]:

$$\delta_{ij} = \frac{\left(k_{ij} - \bar{k_j}\right)}{\bar{k_j}} \tag{1}$$

where $k_{ij}$ is the DGT measured at site *i* and moment *j*.

The mean value of DGT at time *j* is:

$$\bar{k_j} = \frac{1}{m}\sum_i^m k_{ij} \tag{2}$$

where *m* is the number of monitoring wells.

The average relative difference in site *i* is defined as:

$$MRD_i = \frac{1}{n}\sum_{j=1}^n \delta_{ij} \tag{3}$$

where *n* is the number of calculated months.

The standard deviation of the relative difference at site *i* is defined as:

$$SDRD_i = \sqrt{\frac{1}{n-1}\sum_{j=1}^n \left(\delta_{ij} - MRD_i\right)^2} \tag{4}$$

## 3. Results

*3.1. Scale Effects of Spatial and Temporal Variation in DGT*

According to the DGT data in the HID from 1991 to 2015, the annual average DGT changes at different spatial scales are shown in Figure 2. The temporal variation in DGT in the HID was relatively obvious, showing an overall fluctuating downwards trend in Tables S1–S4. Similarly, according to the comparison of the box plots for the four regions at

the annual scale, there was a large difference in DGT in the study area, and the maximum values of DGT in the other three regions were lower than that in the irrigation district scale. Over the 25-year period, the average DGT at the four scales were 1.82 m, 1.76 m, 1.72 m, and 1.79 m, respectively; the average DGT in HID increased from 1.70 m to 2.14 m, with an increasing rate of 2.12 cm/a. The average maximum DGT over the 25 years were 4.96 m, 3.25 m, 2.43 m, and 2.20 m for the four regions, respectively, showing a positive correlation with the change at the regional scale. However, the relationship between the minimum values of DGT and the scales revealed a negative correlation, with average minimum values of 0.74 m, 0.90 m, 1.14 m, and 1.44 m for these four scales, respectively. The minimum depth decreased from 0.85 m to 0.36 m, and the average maximum DGT in the HID increased from 3.06 m to 8.67 m, indicating that the maximum value of DGT in the HID changed more significantly each year. Similarly, with the decrease in the spatial scale, the overall fluctuation range of the characteristic values also decreased. For example, the increasing rate of the maximum depth at the four scales were 0.224 cm/a, 0.103 cm/a, 0.029 cm/a, and 0.010 cm/a. The maximum increasing rate of the minimum depth occurred at the max scale, which was −1.9 cm/a in the HID, and the maximum increasing rate of the average depth was 0.025 cm/a, which occurred at the main canal scale. However, the increasing rate of the minimum DGT at the main canal scale was −0.004 cm/a, as the lowest value, indicating that the maximum value of DGT at the main canal more significantly. Similarly, the increasing rate of the minimum DGT exceeded the rate of maximum and average at the branch canal, respectively, were −28.07%, 11.72%, and 5.65%. It showed that the variation range of DGT was more susceptible to impact at the smaller scales. Based on the variability characteristics of the DGT at the different scales, it can be seen that the number of outlier points reduced as the regional scale decreased, consistent with the change trend of the decreasing maximum rate. This further validates the spatial scale effect mentioned previously, showing different patterns of change in DGT with changing in area size. Therefore, study should focus on DGT in small and medium-sized regions.

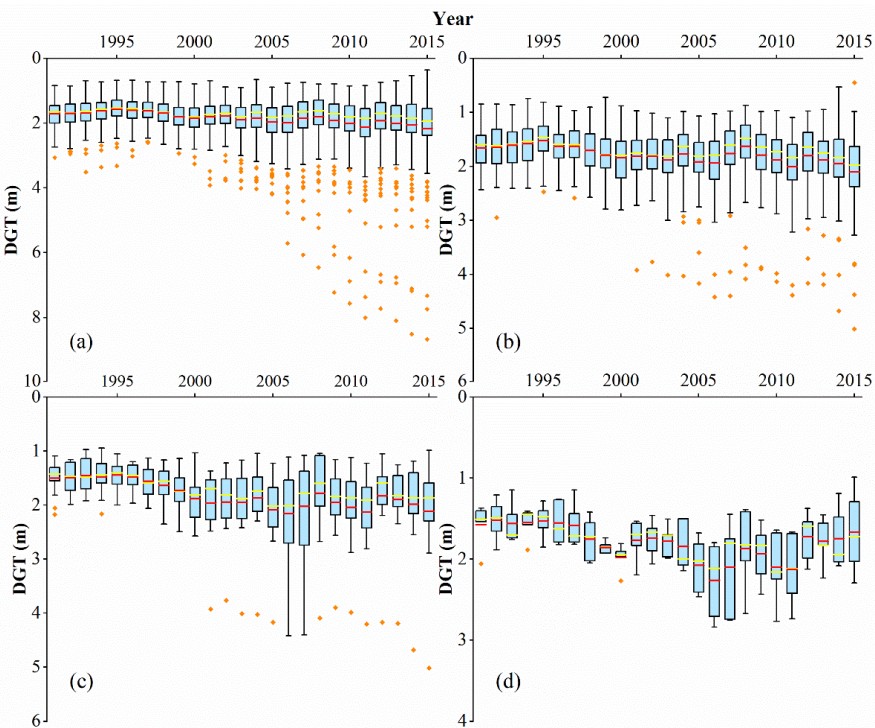

**Figure 2.** Variation characteristics of annual average DGT in the HID at different scales: (**a**) irrigation district scale, (**b**) irrigation subdistrict scale, (**c**) main canal scale, and (**d**) branch canal scale. The yellow line represents the median, the red line represents the mean, and the scattered dots represent the outliers.

According to Figure 3, the DGT had obvious seasonal fluctuations at the four scales, mainly due to dynamic changes in meteorological factors such as precipitation, evaporation, and freeze–thaw cycles. It reached its peak in June and November every year, corresponding to the irrigation seasons in the HID. Usually, irrigation started in early (mid) May, and the amount of irrigation water increased significantly in spring. In addition, soil thawing water replenished groundwater, leading to a gradual decrease in DGT, thereby reaching the first peak with approximately 1.46 m. In autumn, a large amount of irrigation water was applied in a short period to leach salt and maintain soil moisture. As seen from the Figure 3, the DGT rapidly decrease during the autumn irrigation period (from October to November) and reached its minimum of approximately 1.23 m. From December to March of the next year, the DGT gradually increased, and reached its maximum in March, with average maximum values of 2.35 m, 2.29 m, 2.32 m, and 2.35 m for these four scales, respectively. The main reason for this is that the HID begins to freeze after December every year, and the thickness of the frozen soil gradually increases, leading to an increase in DGT. The HID is located in a semiarid region, and the DGT from July to September significantly decreases due to evaporation and crop transpiration, but the DGT was still higher in this period than that during the freezing period.

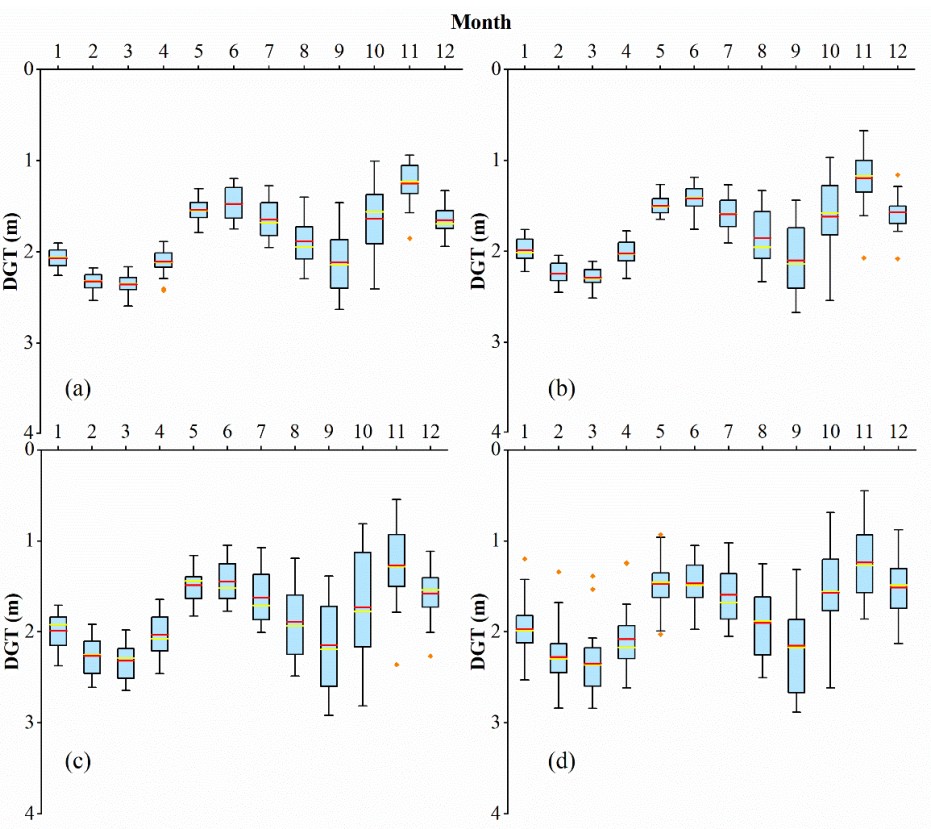

**Figure 3.** Variation characteristics of monthly average DGT in the HID at different scales: (**a**) irrigation district scale, (**b**) irrigation subdistrict scale, (**c**) main canal scale, and (**d**) branch canal scale. The yellow line represents the median, the red line represents the mean, and the scattered dots represent the outliers.

In addition, the coefficients of spatial variation over time were calculated for four different scales: irrigation district scale, irrigation subdistrict scale, main canal scale, and branch canal scale. The 25-year average coefficients of variation for these four scales were 33.22%, 28.63%, 30.02%, and 18.17%, respectively, showing that the coefficient of variation for the main canal scale was larger than that for irrigation subdistrict. By comparing the four sets of data, it was found that the well with the maximum DGT was located in the city covered by the main canal area, and the DGT in this well increased gradually under

the impact of regional groundwater exploitation, which caused the coefficient variation more significant. In order to exclude the influence of this abnormal value, the coefficients of variation for the four scales were calculated after removing it, the average values were 32.85%, 26.61%, 21.58%, and 18.17%, respectively. Obviously, there was a positive correlation between the spatial scales and the coefficients of variation, i.e., the larger the spatial scale was, the larger the corresponding coefficient of variation was (Figure 4). At an annual scale, the overall range of variation coefficients in the HID over 25 years changed from 21.7% in 1991 to 50.0% in 2015, revealing that the changes in DGT became more dispersed with time. To further explore the spatial scale effect of the variation in monthly DGT, the monthly average coefficients of variation for the four scales were calculated as 39.50%, 33.36%, 30.19%, and 27.43%, respectively. By comparing the annual and monthly coefficients of variation, it can be concluded the monthly average values were higher than the annual average regardless of scales, which was due to the alternate impact of meteorological factors and irrigation.

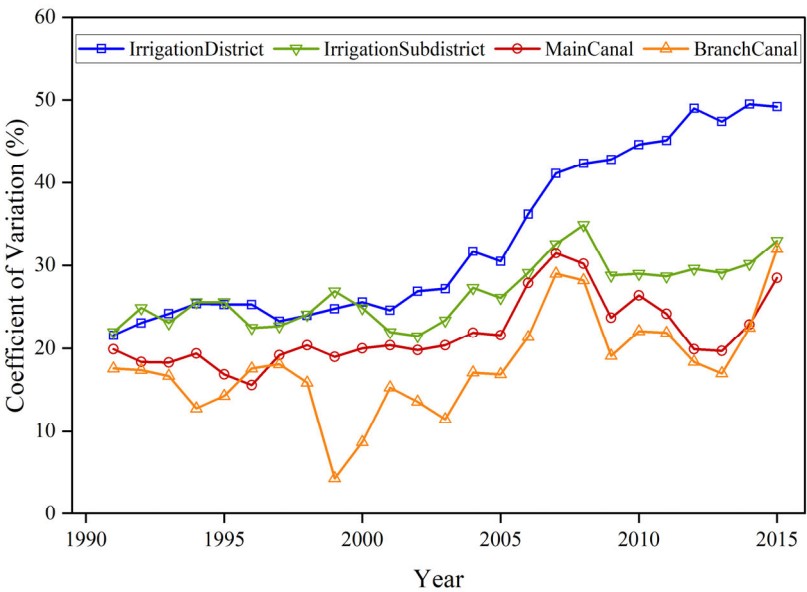

**Figure 4.** Annual coefficients of spatial variation at different regional scales from 1991 to 2015.

Since 1999, in order to improve the utilization rate of water resources, the HID had implemented water-saving projects (WSPs), such as lining canals, leveling farmlands, changing planting structures, etc. [21]. Considering the impact of the application of WSPs on the temporal variability of DGT, this study took 2000 as the time node and explored the changing patterns of the temporal variation coefficients at different scales before and after the implementation of WSPs. First, the time periods were divided into five segments: 1991–1995, 1991–2000, 2001–2005, 2001–2010, and 2001–2015. In each time segment (including before and after the application of WSPs), the spatial scales and monthly variation coefficients were negatively correlated as shown in Table 1. Before the application of WSPs, the variation coefficients of the irrigation district, main canal, and branch canal all decreased with the increase in the time series, while the variation coefficient of the HID kept stable. However, after the implementation of WSPs, the coefficients at the four spatial scales all significantly increased with the increase in the time series, and the smaller the spatial scale was, the more obvious the increase in the variation coefficient.

**Table 1.** Temporal variation coefficients of monthly scale in different regions from 1991 to 2015.

|           | Irrigation District | Irrigation Subdistrict | Main Canal | Branch Canal |
| --------- | ------------------- | ---------------------- | ---------- | ------------ |
| 1991–1995 | 31.49%              | 32.19%                 | 34.29%     | 36.27%       |
| 1991–2000 | 31.58%              | 30.61%                 | 32.89%     | 33.15%       |
| 2001–2005 | 28.00%              | 27.25%                 | 28.24%     | 30.73%       |
| 2001–2010 | 30.06%              | 30.23%                 | 30.76%     | 32.45%       |
| 2001–2015 | 31.21%              | 31.38%                 | 32.66%     | 35.00%       |

*3.2. Changes in the Area of Different Ranges of DGT*

The spatial interpolation of DGT in the HID was carried out by using the inverse distance interpolation method with a 5-year interval as shown in Figure 5. The results showed that change in the DGT ranged from 0.7 to 3.1 m in the study area, and the spatial distribution of DGT in the HID was deeper in the north and south and shallower in the middle, showing tremendous regional variability. Although there was a slight recovery trend of DGT in some regions from 2006 to 2010, it was still unable to regain the depth before 2000. The areas with deeper DGT increased significantly over time. Furthermore, to explore the differences in DGT at different spatial scales, the area ratio occupied by DGT in four ranges were also calculated shown in Figure 6. According to the variation in DGT in the HID, it was divided into four levels: 0.5 < DGT ≤ 1.5 m, 1.5 < DGT ≤ 2.5 m, 2.5 < DGT ≤ 3.5 m, and >3.5 m, representing shallow, moderate, deep, and extremely deep depths, respectively. The period from 1991 to 2000 was taken as the period before the application of WSPs, and every 5a thereafter was taken as the period after the application of WSPs; respectively, the initial stage (2001–2005), the middle stage (2006–2010) and the late stage (2011–2015). For four periods, the area ratio occupied by DGT in the 1.5–2.5 m range was the largest, accounting for 94.11%, 96.40%, 86.13%, and 79.43%, respectively, showing first increase and then decrease. This indicated that the change in the DGT in the HID mainly ranged from 1.5 to 2.5 m, which was beneficial to control the soil salinization. The ratio of depths in the 0.5–1.5 m range decreased over time, from 5.86% to 2.78%. Meanwhile, the areas occupied by depths in the 2.5 < DGT ≤ 3.5 m and >3.5 m ranges increased annually, the former increased from 0.03% to 15.66% and the latter only appeared after the application of WSPs, accounting for 0.02%, 1.04% and 2.13%.

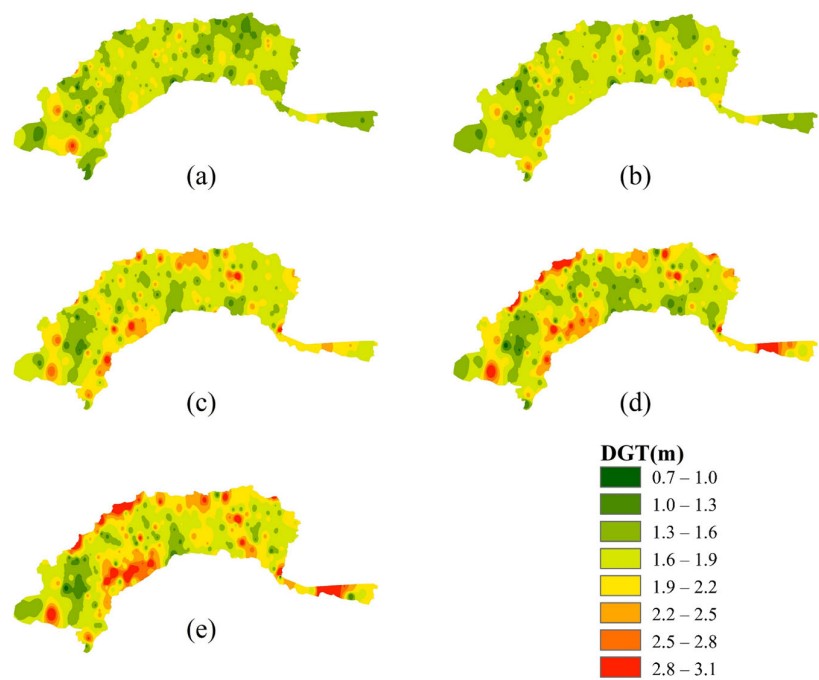

**Figure 5.** Spatial distribution of the 5-year average DGT in the HID in (**a**) 1991–1995, (**b**) 1996–2000, (**c**) 2001–2005, (**d**) 2006–2010, and (**e**) 2011–2015.

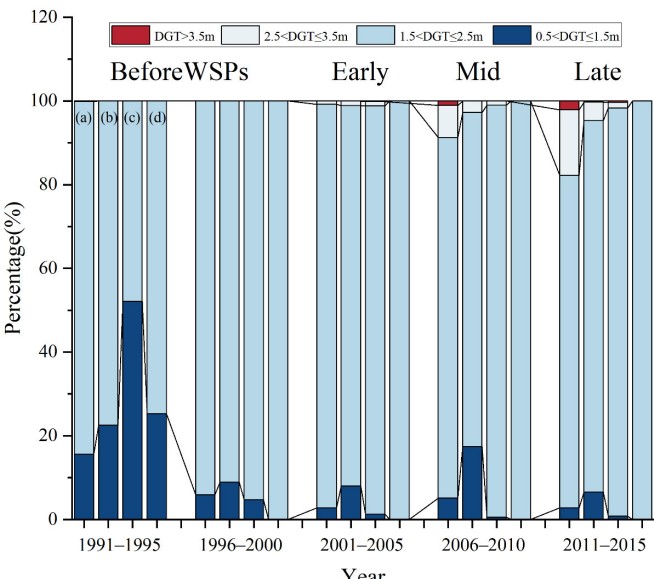

**Figure 6.** Area change of 5-year interval of DGT in the HID at different scales: (**a**) irrigation district scale, (**b**) irrigation subdistrict scale, (**c**) main canal scale, and (**d**) branch canal scale.

At the four scales, irrigation district area, irrigation subdistrict area, main canal, and branch canal, there were significant differences in the DGT. However, all four scales showed strong consistency: the area occupied by DGT in the 1.5–2.5 m range was the largest. Especially at the branch canal scale, the area occupied by depths in the 1.5–2.5 m range could be as high as 100%. The HID had a large area and different planting structure, and the irrigation water use efficiency showed a downward trend from the upper reaches to the lower reaches. Therefore, it was significantly influenced by other regions, and the scale effect was limited by the regional scope. Except for the irrigation district scale, the average area ratio in the 1.5–2.5 m range over the 25 years were 85.64%, 87.35%, and 94.95% for the other three regions, respectively, showing a negative correlation with the change in the regional scale. However, the average area ratios in the 1.5–2.5 m range were 84.33%, 71.63%, and 87.38% before the application of WSPs, which were lower than these after the application of WSPs, 86.51%, 97.84%, and 100.00%, respectively. Therefore, it showed that implementing the WSPs had positive effect on the 1.5–2.5 m range, but the area of deep DGT was still increasing.

### 3.3. Analysis of Temporal Stability in DGT

3.3.1. Autocorrelation Analysis of DGT

The main purpose of temporal stability analysis is to accurately estimate the average DGT at a given site. The Spearman rank correlation coefficient method was used to evaluate the similarity of spatial patterns of monitoring sites over time. As shown in Figure 7, in most years, these relationships were significant at $p < 0.01$ or $p < 0.05$, indicating that the spatial patterns of DGT in the HID had strong temporal persistence. From the Figure 7, it can be seen that the correlation among annual DGT showed a significant separation in 2000. The average correlation coefficients were 0.85 and 0.93 between adjacent two years before and after the application of WSPs. That is, the DGT from 2001 to 2015 had a stronger correlation. This was due to the application of WSPs since 2000, which had a significant impact on changes in DGT. Many studies have also confirmed this view [21,31,32]. Therefore, further analysis of the temporal stability of DGT in the HID needs to be carried out to explore stability sites before and after the project.

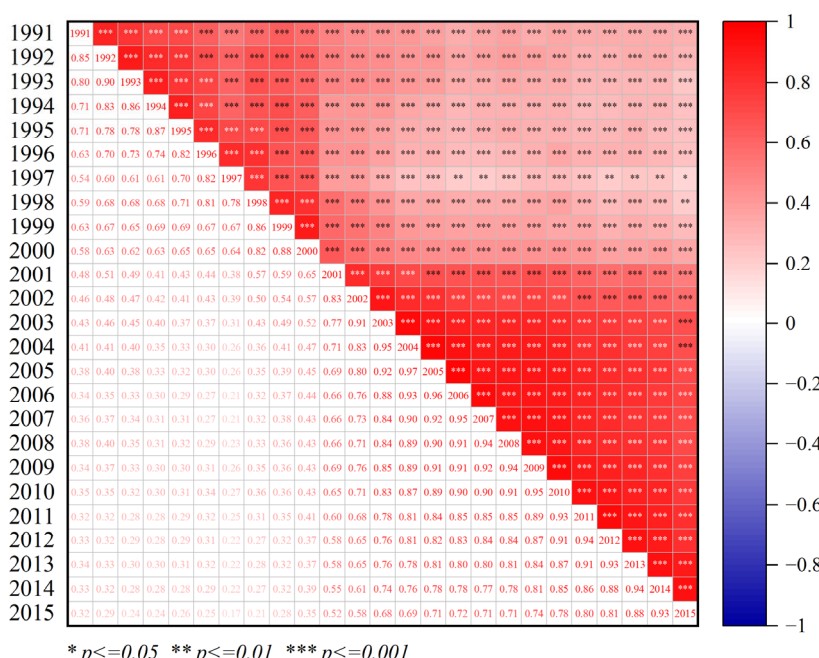

*\* p<=0.05  \*\* p<=0.01  \*\*\* p<=0.001*

**Figure 7.** Spearman grade correlation coefficient of the HID over 25 years.

### 3.3.2. Identification of Stable Monitoring Sites

Based on the previous analysis, we took 2000 as a time node and divided the period from 1991 to 2015 into two stages: before the application of WSPs (1991–2000) (Figure 8) and after the application of WSPs (2001–2015) (Figure 9). The three different time scales were considered, i.e., 15-year, 10-year, and 5-year, respectively. The mean relative difference (MRD) and standard deviation relative difference (SDRD) of each observation well were determined, and the MRD ascending order results are shown in Figures 8 and 9. Some studies have reported that when the SDRD is less than 35%, the DGT is relatively stable over time [30,33]. In this study, the SDRD values were controlled to be less than 30% and the MRD values were less than 0.1, which can make the selection of stable sites more accurately.

The results showed that the stability sites before and after the application of WSPs were not entirely the same as those obtained from 1991 to 2015. In addition, the stability sites appeared with different frequencies for three temporal scales (15-year, 10-year, and 5-year). Before the implementation of WSPs, the stability sites were W90, W98, W167, W229, and W302 (Figure 8). After the implementation of WSPs, especially at the 15-year scale, the stability sites were W193, W229, W302, W222, W267, W51, and W215 (Figure 9c). W229 and W302 also occurred at 10-year and 5-year scales as stability sites (Figure 9a,b). It can be inferred that both W229 and W302 showed good temporal stability at a long-term scale before and after the application of WSPs.

Before and after the application of WSPs, the spatial distribution of the stability sites also showed significant differences. Before the application of WSPs, the stability sites were mainly concentrated in the central part of the HID, showing an eastward shift trend with the increase in time scale (Figure 10). After the application of WSPs, the stability sites were mainly located in the central part of the HID at a 5-year scale. When the time scale increased, the stability sites were mainly distributed in the eastern part of the HID (relatively concentrated within the Yichang Irrigation Subdistrict). However, it was worth noting that W229 and W302 showed significant stability before and after the application of WSPs. In conclusion, W302 and W229 were considered the most representative sites during the long-term scales. Therefore, the application of WSPs significantly affected the temporal stability of DGT, and although the W229 and W302 sites were affected to some extent, they quickly restored stability, making it more representative of the region. Considering most of the DGT in the HID were within a range of 1.5–2.5 m, the average depth of

W229 and W302 over 25 years were 1.81 m and 1.82 m, which considered to be of medium DGT, indicating that it had better time persistence. Among the stability sites mentioned above, the DGT was basically stable at approximately 2 m. This indicated that the area with DGT between 1.5 and 2.5 m had strong time stability in the HID, while the DGT exceeded this range with poor time stability. Therefore, the GDT measurement in Yichang Irrigation Subdistrict will effectively achieve dynamic monitoring of DGT in the HID.

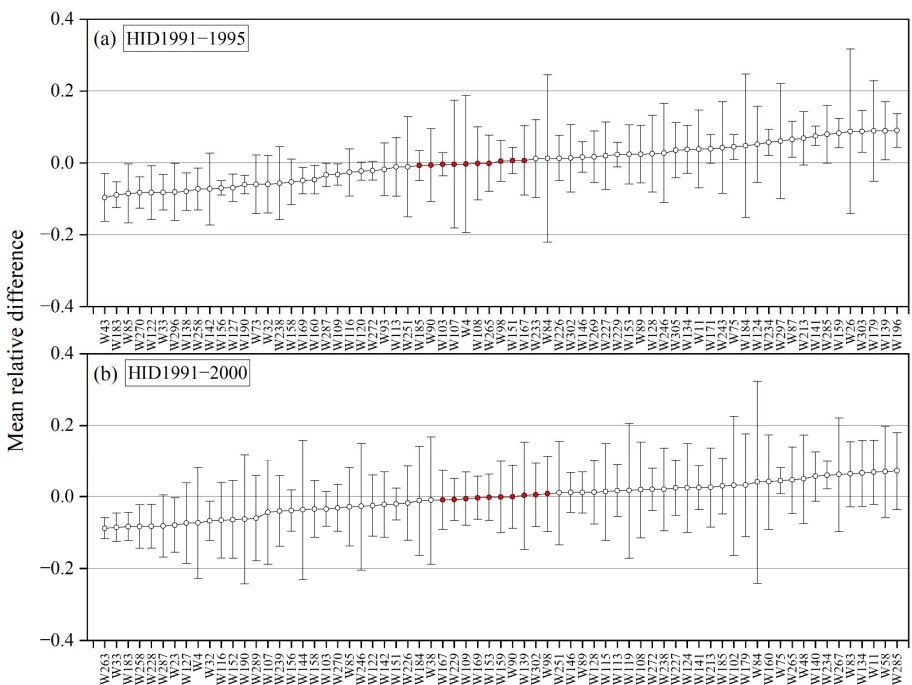

**Figure 8.** MRD of monitoring wells in the HID (**a**) from 1991 to 1995 and (**b**) from 1991 to 2000, the error line represents SDRD, and red points are wells with stability.

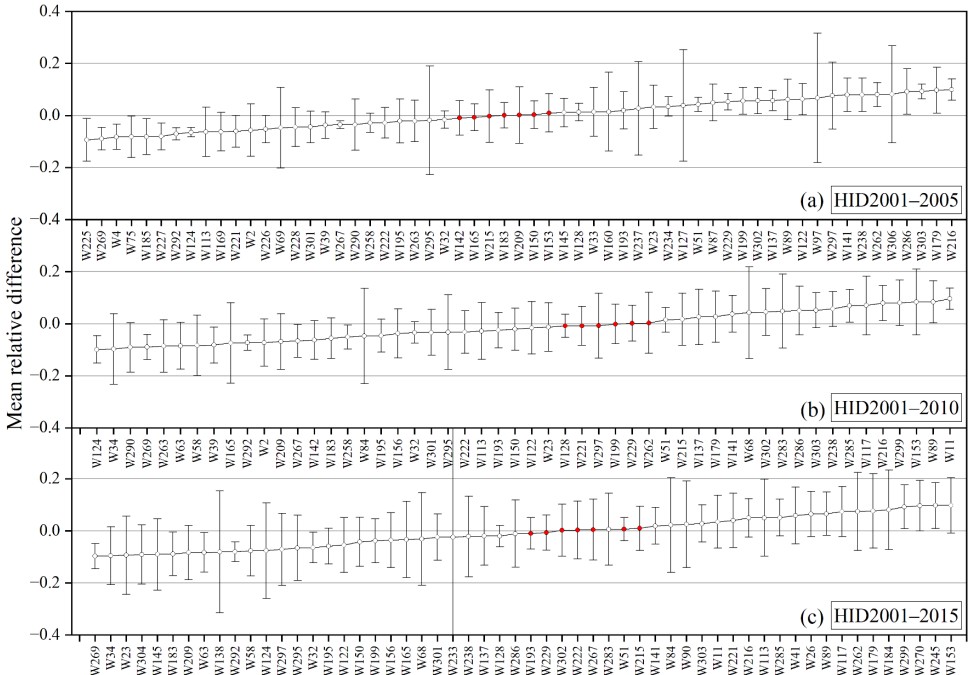

**Figure 9.** MRD of monitoring wells in the HID (**a**) from 2001 to 2005, (**b**) from 2001 to 2010 and (**c**) from 2001 to 2015, the error line represents SDRD, and red points are wells with stability.

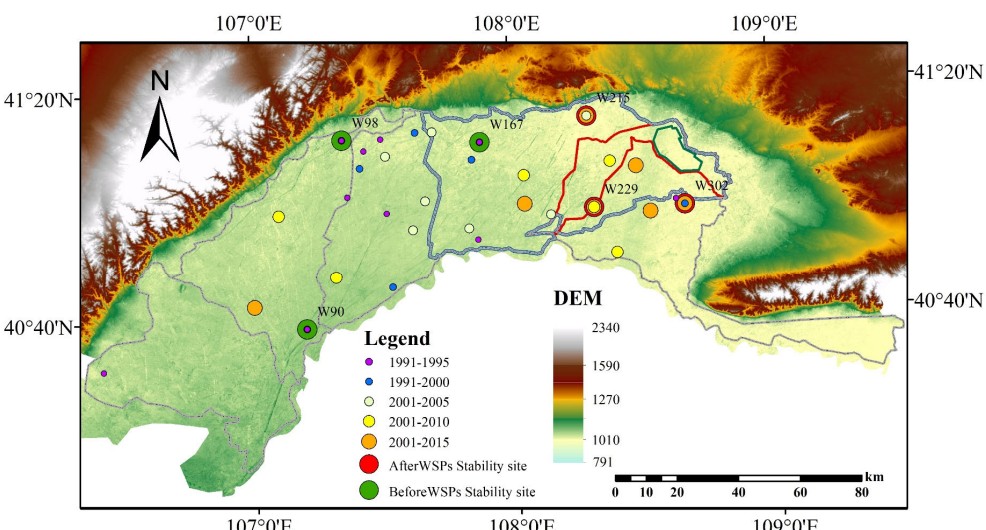

**Figure 10.** Before and after the application of WSPs stability sites of DGT in the HID.

### 3.3.3. Rationality Test of Stability Sites

To determine whether representative sites can accurately estimate the average DGT in the HID, the DGT of representative wells measured before and after the implementation of WSPs were compared with the average DGT of all observation wells in the HID. As shown in Figure 11, the correlation between the DGT of stability sites before and after the application of WSPs and the average DGT in the HID was compared. The results showed that the representativeness of the stability sites was affected after implementing WSPs, and the correlation was significantly reduced, but they were still representative. As mentioned, the stability sites were W229 and W302, but after implementing WSPs, W229 had an advantage in representativeness. The DGT of well W229 fluctuated near the average value, but the $R^2$ value after implementing WSPs was only 0.576. It also indicated that the application-of-WSPs process will have a significant impact on the DGT of different wells, especially the stability sites. Compared to the average DGT of W229 and the regional average values before and after the application of WSPs, the errors were approximately 0.015 m/m and 0.013 m/m, respectively. Although the stability sites that only occurred before implementing WSPs such as W90 and W167, showed good correlation, which $R^2$ were 0.769 and 0.615, their fitting results after the application of WSPs were much lower than the other stability sites mentioned above. This also indicated that the other stability sites can still maintain their stability advantage even when they are interfered with external factors.

According to the above analysis, this article selected W229 as the optimal stability sites. It has shown good stability characteristics both before and after the application of WSPs, with a high correlation with the average DGT in the HID. This article suggested that in the future groundwater monitoring process, monitoring of the above sites in the Yichang Irrigation subdistrict and other areas should be strengthened to avoid resource waste during the monitoring process.

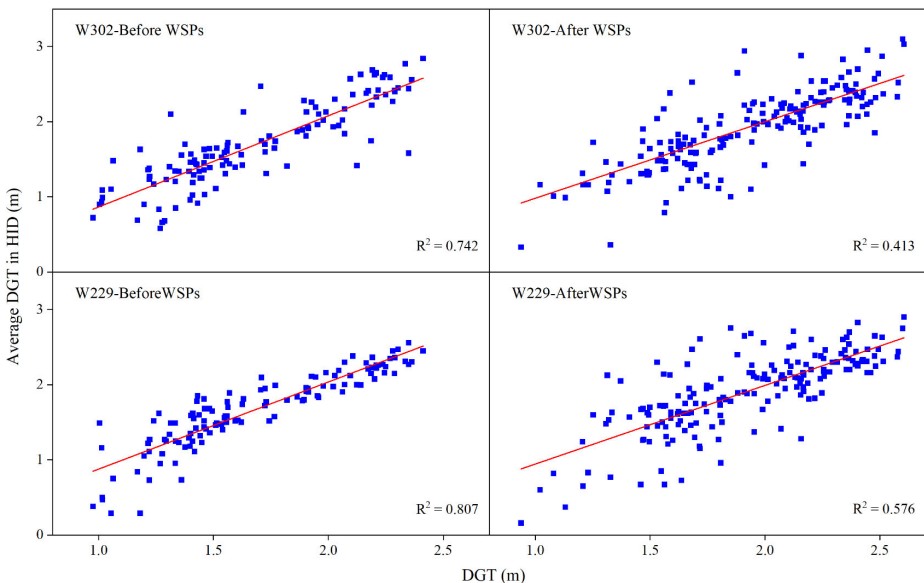

**Figure 11.** Correlation between the DGT of representative sites and the average DGT of the irrigation area before and after the application of WSPs.

## 4. Discussion

In this study, the area with DGT ranging from 0.5 to 1.5 m accounted for 10% of the total irrigation area from 1991 to 2000. However, starting from 1996, the area with this depth began to decrease, while the area with DGT greater than 2.5 m increased, but the overall trend remained dominated by moderate DGT (1.5–2.5 m). This was consistent with the findings of Yue et al. [19] in the Yichang Irrigation Area (the largest irrigation subdistrict of the HID), which showed an increase in the area of DGT ranging from 1.5 to 2.5 m and 2.5–3.5 m. Meanwhile, the DGT strongly followed the seasonal characteristics of precipitation and irrigation in terms of spatial and temporal patterns. This was consistent with the results of Yuan et al. [34] and their experiments on regional climate models, especially in semiarid and humid China, and there was a close relationship between groundwater level dynamics and precipitation changes. For the intraannual fluctuations of DGT, the effect of irrigation was more significant than that of evaporation on DGT in June and November, which was also validated by Yue et al. [19]. In addition, meteorological factors were not only the key factors affecting the variation in DGT, but also constrained the spatiotemporal variation of soil moisture [35]. Moreover, the relevant studies have shown that climate change changes the spatiotemporal variability of groundwater through its impact on groundwater recharge [21,36] and the DGT decreased during wet periods and increased during dry periods [37].

In terms of scale effects, this study found that external factors, especially the application of WSPs, significantly affected the variability and range of DGT at different scales, with more sensitive effects at small and medium scales. This is consistent with the findings of Lu et al. [33], who found that the selection of sampling sites at small and medium scales tended to make soil salinity more homogeneous at small scales and more heterogeneous and complex at large scales. When exploring the spatiotemporal variability characteristics of DGT, it was found that as the area scale decreased, the spatial variation coefficient of DGT also decreased, but it remained moderately variable and changed similarly at different scales. Spearman's rank correlation coefficient and the correlation analysis with time lag showed that the spatial pattern of DGT remained similar during a certain period, and the similarity decreased as the period increased. Similar results were found in other regions regarding DGT and soil moisture. For example, Wang et al. [30] and Schneider et al. [38] found that the spatial pattern of soil moisture content remained similar within 2–3 years, while the similarity of DGT in the Yellow River Delta region persisted within 1.5–3 years and decreased with increasing time intervals. Comparison revealed that the persistence of

similarity in the HID can last for 10–15 years, much longer than in other regions, which may be closely related to local water-saving policies. This may also be related to the amount of data. If the data are enough, the continuous similarity of DGT will increase accordingly. Although this study did not reveal the spatial and temporal correlations of DGT, such correlations are likely to exist, and research has shown that the relative standard deviation in space was approximately three times than that in time [39].

Over time, the stability sites of DGT have changed under the application of WSPs. In exploring the temporal stability of soil water, Diego Rivera et al. [40] believed that the change in stability sites mainly depended on the amount of water entering the soil, as well as the previous soil moisture content. In this article, the stability sites changed more significantly before and after the application of WSPs, and the main reasons for the change were the reducing irrigation infiltration and the increasing groundwater evaporation [3]. For example, Xu et al. [41] used MODFLOW and GIS to announce that the application of WSPs could save groundwater by reducing evaporation. Yue et al. [21] used the wavelet analysis method to find that the driving factor of the change in DGT before and after the implementation of WSPs has shifted from meteorological factors to irrigation factors. The findings they obtained suggest that the links between groundwater and the WSPs can be intricate and possibly alternating. In the future work, more comprehensive water-saving measures should be considered from the above perspectives to ensure the rational planning and utilization of water resources.

## 5. Conclusions

In this study, based on the monthly data of 206 monitoring wells in the HID from 1991 to 2015, the results showed that the variation coefficient of the HID was positively correlated with the reduction in spatial area at different regional scales. The average annual variation coefficients of the four regional scales are 32.85%, 26.61%, 21.58%, and 18.17%, which are lower than the average monthly variation coefficients (39.82%, 32.19%, 37.52%, and 27.43%), but all belonged to medium variation. The results of Spearman rank correlation analysis showed that the spatial pattern of DGT was correlated in a certain period, and the correlation was weakened beyond this period (10–15 years). In addition, this study also found that the interannual correlation of DGT before the implementation of WSPs (1991–2000) was higher than that after the implementation of WSPs (2001–2015). It was believed that implementing WSPs could affect the temporal stability of DGT and reduce the representativeness of stable sites. This was due to a reduction in infiltration recharge caused by implementing WSPs, resulting in greater fluctuations in groundwater levels. Moreover, the stability of monitoring wells with medium DGT was higher than that of wells with shallow and deep DGT. Although the DGT of representative monitoring well (W229) has a strong time stability mode, the proportion of stability sites was 9.7%, indicating that the DGT in the HID had significant spatiotemporal variation characteristics, and the variation in the west was greater than that in the east of the HID. Overall, the accurate identification of the temporal stability sites of DGT in HID based on dynamic changes in groundwater level provides useful information for water resource management, and the long-term monitoring of stable sites can evaluate the change in the average DGT, which will be of great benefit to the adjustment of WSPs in time. At the same time, understanding different vegetation cover types and terrains, as well as the impact of climate change on groundwater stability, needs to be considered in the future work to serve long-term water resource planning. This study can provide a theoretic basis for the intensive economic utilization of water resources, and the improvement of soil salinization.

**Supplementary Materials:** The following supporting information can be downloaded at: https://www.mdpi.com/article/10.3390/agronomy13082172/s1, Table S1. Statistical characteristic value of Hetao Irrigation District from 1991 to 2015. Table S2. Statistical characteristic value of Yichang Irrigation Subdistrict from 1991 to 2015. Table S3. Statistical characteristic value of Yihe Main Canal from 1991 to 2015. Table S4. Statistical characteristic value of Branch Canal from 1991 to 2015.

**Author Contributions:** Z.Z.: conceptualization, investigation, and writing—original draft preparation. W.Y.: methodology, investigation, formal analysis, validation, project administration, and funding acquisition. H.Z.: investigation, review, and revisions. C.C.: writing—original draft and review. F.W. and P.L.: software, validation, supervision, and data curation. J.W.: improvement of research ideas, visualization, and revision of manuscript. All authors have read and agreed to the published version of the manuscript.

**Funding:** This study was funded by the National Key Research and Development Program of China (Grant No. 2021YFC3201204), the National Natural Science Foundation of China (Grant No. 52179032 & 51879011), and the Program of Monitoring of Groundwater and Major Lakes along the Yangtze River in Hubei Province (420000-2023-218-006-003).

**Data Availability Statement:** Not applicable.

**Acknowledgments:** The authors are grateful for the support from the Experimental Station of Yichang Sub-center of Water Development Center of Hetao Irrigation Dsitrict in Inner Mongolia.

**Conflicts of Interest:** The authors declare no conflict of interest.

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
