# Peer review of "Analysis of the Scale Effect and Temporal Stability of Groundwater in a Large Irrigation District in Northwest China"

_agronomy, doi:10.3390/agronomy13082172_

Round 1
Reviewer 1 Report
The manuscript entitled “Analysis of the scale effect and temporal stability of groundwater in a large irrigation district in Northwest China” is mainly based on the study carried out from the groundwater management perspective. The authors have tried to highlight their study based on the previous data and they have analyzed the data in a good manner. The manuscript is written well and the representation of the work is good enough. From my side, the current version of the manuscript is in good format and may be considered for possible publication. Some typing mistakes I have observed in the manuscript. Hence authors are requested to check the whole manuscript once again.
The English language used in the manuscript is good and can be easily understood.
Reviewer 2 Report
I have had the opportunity to carefully review the manuscript titled "Analysis of the scale effect and temporal stability of groundwater in a large irrigation district in Northwest China" and would like to commend the authors on their insightful and well-structured study. The paper presents valuable findings regarding the spatiotemporal dynamics of Depth to Groundwater (DGT) in the studied region and makes a significant contribution to the field of hydrology and water resource management.
The authors have done an excellent job in conducting a thorough analysis of the DGT data from 206 monitoring wells over a period of 25 years (1991-2015). The research methodology is well-structured, and the statistical analysis utilized is appropriate and robust. The figures and tables included in the manuscript are clear and effectively support the presented findings.
The paper effectively discusses the variation in DGT at different spatial scales, emphasizing the correlation between variation coefficients and the reduction in spatial area. The study's focus on the seasonal fluctuations of DGT and the impact of regional climate and human activities on groundwater levels adds significant value to the research.
The investigation into the temporal stability of DGT, using Spearman rank correlation analysis, provides valuable insights into the persistence of spatial patterns over time. The authors' identification of stability sites and their evaluation of the impact of Water-saving Projects (WSPs) on temporal stability enhance the paper's significance.
In light of the above strengths, I believe that the manuscript is well-written and presents a coherent and valuable contribution to the literature. I have only a few suggestions for minor adjustments that would further enhance the paper:
Provide a clearer explanation of the implications of the findings for agricultural production and resource management in the study area.
Consider expanding the discussion on the potential impact of climate change on groundwater dynamics and how it may affect the study's results and conclusions.
Incorporate more context regarding the impact of land cover types and terrain on the observed spatiotemporal variation.
Overall, I believe that these minor adjustments will strengthen the manuscript further and ensure that it makes an even more significant impact in the field.
Moreover, some other issues are the following:
Line 73: discuss geostatistics and GIS technology, it would be helpful to provide a brief explanation of what GIS technology stands for and how it relates to the analysis of spatiotemporal variability. This addition can provide more context for readers who might not be familiar with GIS.
Line 112: the mention of "Water-saving practices (WSPs)" as a factor impacting temporal stability could benefit from a brief explanation or clarification. What specific water-saving practices are being referred to here? Providing a sentence or two to explain this term would enhance the reader's understanding.
Line 159: The text mentions that there was an overall fluctuating downwards trend in DGT, but it would be beneficial to include specific statistical analysis or trends over time to support this observation. For instance, you can include regression lines or statistical significance of the trend.
Reviewer 3 Report
Paper "Analysis of the scale effect and temporal stability of groundwater in a large irrigation district in Northwest China" presents research related to the hydrogeology. I give the following suggestions related to scientific paper:
- Replace certain keywords so that they do not match the words in the title. Select them from the abstract.
- The abstract should not contain polemics about the results. The starting hypothesis and methodology for solving the problem should be briefly written here.
- A lot of literature is related to study areas in China. It is proposed to include problems from all over the world.
- Figure 1 is not so representative. As stated even in the title, the problem of groundwater is considered. Therefore, a hydrogeological map and a hydrogeological profile are proposed.
- The methodology is described very briefly. Some equations are shown. A more detailed presentation of applied scientific methods is suggested. There is no call for references, so it seems as if the authors of the scientific paper are also the authors of some well-known statistical equations.
- Statistical parameters are not adequately displayed. It is not representative to list their values in the text. A diagram and a tabular presentation are suggested.
- Figure 5 is not representative. It is necessary to add another layer, at least a topographical map.
- Figure 7 is really unrepresentative. The values displayed in this way have very poor visibility and are not clear. Here is an example of a scientific paper where statistical parameters and coefficients were processed and presented in a very interesting way: Kresojević, M.; Ristić Vakanjac, V.; Trifković, D.; Nikolić, J.; Vakanjac, B.; Polomčić, D.; Bajić, D. The Effect of Gravel and Sand Mining on Groundwater and Surface Water Regimes - A Case Study of the Velika Morava River, Serbia. Water 2023, 15, 2654. https://doi.org/10.3390/w15142654
- Chapter 4 should be merged with chapter 3. There is no need to separate it.
- The conclusion should be revised. It should not take the form of an summary. The scientific contribution as well as the contribution to the academic community should be emphasized here.
